# ADP-Ribosylation in Antiviral Innate Immune Response

**DOI:** 10.3390/pathogens12020303

**Published:** 2023-02-12

**Authors:** Qian Du, Ying Miao, Wei He, Hui Zheng

**Affiliations:** 1Institutes of Biology and Medical Sciences, Soochow University, Suzhou 215123, China; 2Jiangsu Key Laboratory of Infection and Immunity, Soochow University, Suzhou 215123, China

**Keywords:** ADP-ribosylation, PARylation, MARylation, post-translational modifications, viral infection, antiviral response, IFN-I, innate immunity

## Abstract

Adenosine diphosphate (ADP)-ribosylation is a reversible post-translational modification catalyzed by ADP-ribosyltransferases (ARTs). ARTs transfer one or more ADP-ribose from nicotinamide adenine dinucleotide (NAD^+^) to the target substrate and release the nicotinamide (Nam). Accordingly, it comes in two forms: mono-ADP-ribosylation (MARylation) and poly-ADP-ribosylation (PARylation). ADP-ribosylation plays important roles in many biological processes, such as DNA damage repair, gene regulation, and energy metabolism. Emerging evidence demonstrates that ADP-ribosylation is implicated in host antiviral immune activity. Here, we summarize and discuss ADP-ribosylation modifications that occur on both host and viral proteins and their roles in host antiviral response.

## 1. Introduction

Rapid and appropriate cellular responses are essential for organisms to respond to external stimuli. Post-translational modifications (PTMs) play an important role in this process by modulating intracellular signal transduction pathways [1]. Mechanically, PTMs regulate signaling pathways and gene expression mainly by affecting the catalytic activity of the target protein or the interaction of the target protein with other molecules [2]. Adenosine diphosphate (ADP)-ribosylation is an ancient PTM discovered around the 1960s [3]. ADP-ribosyltransferases (ARTs) are responsible for ADP-ribosylation [4]. ARTs are a superfamily consisting of 23 members, two of which are diphtheria toxin-like ADP-ribosyltransferase (ARTD) and cholera-like ADP-ribosyltransferase (ARTC) [5]. The ARTD family includes PARPs and TNKS (tankyrase), as detailed in Table 1. ARTs transfer one or more ADP-ribose (ADPr) units from nicotinamide adenine dinucleotide (NAD^+^) to target proteins on a variety of amino acids, including lysine (Lys), arginine (Arg), glutamate (Glu), aspartate (Asp), and cysteine (Cys) [6], leading to mono-ADP-ribosylation (MARylation) or poly-ADP-ribosylation (PARylation) of substrates. The human genome encodes 17 PARPs [7], most of which share a common NAD^+^ binding motif in their catalytic domain with a similar secondary structure [8,9]. In these NAD^+^ binding motifs, a histidine (His) residue and a tyrosine (Tyr) residue are essential for positioning NAD^+^ in a correct orientation, and a conserved Glu residue is crucial for ADP-ribose transfer. The histidine-tyrosine-glutamate motif (H–Y–E motif and variants thereof) is also known as the catalytic triad of PARPs [10,11]. However, not all PARPs reserve these critical residues. For example, PARP13 lacks catalytic activity due to the substitution of a His residue on the key NAD^+^-binding motif. Therefore, PARP13 is the only PARP family member with a catalytically inactive domain [12,13]. In fact, only PARP1, 2, 3, TNKS1 and TNKS2 retain these residues and catalytic PARylation [14], while the rest of PARPs (except PARP13) possess the MARylation activity due to the substitution of the Glu residue [8,15,16]. In brief, the ability of PARPs to catalyze MARylation or PARylation of their protein substrates depends on conserved structural features such as the catalytic triad and the presence of certain cofactors.

As for degradation of the poly-ADP-ribose chain, poly-ADP-ribose glycohydrolase (PARG) and ADP-ribosyl-acceptor hydrolase (ARH) 3 are responsible for it. PARG can only hydrolyze ribose-ribosyl *O*-glycosidic bonds, and the final product of hydrolysis is a MARylated protein. While ARH3 can hydrolyze the bond between the amino acid side chain and the ribose, thereby completely reversing ADP-ribosylation. The chemical bond between the amino acid side chain and ribose is different from the ribose-ribose bond in the PAR chain, which explains why different enzymes are required to accomplish the two reactions. Moreover, ARH1 is also an ADP-ribosylhydrolase, which can specifically hydrolyze MARylated proteins at arginine residues [17,18]. In addition, three macrodomain-containing enzymes, MacroD1, MacroD2, and TARG1, can also act as mono-ADP-ribosylhydrolases [19,20,21]. Notably, ARH3 acts primarily at the terminal sites of the PAR chains and the serine residues of MARylated proteins, while MacroD1, D2, and TARG1 are involved in hydrolyzing ADP-ribosylation of acidic residues [2]. In a word, ADP-ribosylation is a fully reversible PTM. The dynamic balance of ADP-ribosylation synthesis and degradation provides plasticity for rapid cellular response to external signals.

ADP-ribosylation occurs in both prokaryotes and eukaryotes and is particularly prevalent in stress responses requiring rapid adaptation [22,23]. MARylation and PARylation are involved in a variety of physiological and pathophysiological processes, including DNA damage recognition, chromatin regulation, and oxidative stress response [24,25,26,27,28,29,30]. During this last decade, several studies uncovered the roles of ADP-ribosylation in antiviral immunity [31,32]. Of note is that there are many PARPs that perform antiviral functions without their catalytic activity, and will not be discussed in detail here [15]. In this review, we mainly summarize and discuss new findings on the role of ADP-ribosylation occurred on the host and viral proteins in antiviral response. As to other substrates, such as nucleotides or other small molecules, we will only give a brief introduction here (Table 1).

**Table 1 pathogens-12-00303-t001:** The enzymes that catalyze ADP-ribosylation and their main functions in antiviral innate immune response.

Name	ADP- Ribosylation Activity	Other Names	Roles in Antiviral Innate Immune Response
**PARP1**	PARylation	ARTD1	PARylating EBNA1 and LANA [33,34] Degradation of IFNAR1 [35]
**PARP2**	PARylation	ARTD2	Not known
**PARP3**	PARylation	ARTD3	Regulating PARP1 [31]
**PARP4**	MARylation	ARTD4	Not known
**PARP5a**	PARylation	ARTD5 Tankyrase 1	PARylating MAVS and promoting its degradation [36] PARylating EBNA1 [37]
**PARP5b**	PARylation	ARTD6 Tankyrase 2	PARylating MAVS and promoting its degradation [36] PARylating EBNA1 [37]
**PARP6**	MARylation	ARTD17	Not known
**PARP7**	MARylation	ARTD14, BAL3	MARylating and inhibiting TBK1 [38] MARylating PARP13 [23]
**PARP8**	MARylation	ARTD16	Not known
**PARP9**	MARylation	ARTD9, BAL1	Forming a complex with DTX3L and MARylating ubiquitin at Gly76 [39] Suppressing PARP14-mediated MARylation of STAT1 [40]
**PARP10**	MARylation	ARTD10	MARylating nsP2 of CHIKV and promoting its degradation [41]
**PARP11**	MARylation	ARTD11	Targeting β-TrCP and promoting IFNAR1 degradation [42] Enhancing PARP12-mediated NS1 and NS3 ADP-ribosylation [43]
**PARP12**	MARylation	ARTD12	Targeting NS1 and NS3 of ZIKV and promoting its degradation [44]
**PARP13**	Inactive	ARTD13	Restricts viral replication [45,46,47,48]
**PARP14**	MARylation	ARTD8	MARylating STAT1 and preventing its phosphorylation [40] MARylating PARP13 [49]
**PARP15**	MARylation	ARTD7	Not known
**PARP16**	MARylation	ARTD15	Not known

## 2. ADP-Ribosylation of Host Proteins

The innate immune system is the main defense mechanism of higher organisms against pathogens such as viruses. It senses and responds to pathogen-associated molecular patterns (PAMPs) through pattern recognition receptors (PRRs). During this process, interferons (IFNs) are produced [50,51,52], enabling a rapid host response to viral infection.

Interferons are commonly classified into three types according to their receptor complex, designated types I to III [53,54]. Type I IFN (IFN-I) comprises IFNα and IFNβ, and most virus-infected cells are able to produce IFN-I to resist viruses. Type II IFN (IFN-II) consists only of IFN-γ, which is synthesized by certain immune cells. However, it can offer resistance to a wide range of pathogens [55]. Type III IFN (IFN-III) includes IFN-λ, which can be produced by most cell types. IFN-III plays an important role in the innate immune response of the intestinal and respiratory mucosal barriers [56,57,58]. Among the three types of IFNs, IFN-I is well characterized and plays an important role in the host response against viral infection [59,60,61]. After IFN-I is produced, it first binds to its membrane receptors (IFNAR1 and IFNAR2) to induce their dimerization and then initiates the autophosphorylation of Janus kinase 1 (JAK1) and tyrosine kinase 2 (TYK2). Phosphorylated JAK1 and TYK2 recruit and activate the signal transducers and activators of transcription 1 (STAT1) and STAT2. Phosphorylated STAT1 and STAT2, together with IFN-regulatory factor 9 (IRF9), form a well-characterized complex, IFN-stimulated gene factor 3 (ISGF3). Then, ISGF3 translocates to the nucleus and binds to IFN-stimulated response elements (ISRE), the promoter of IFN-stimulated genes (ISGs), leading to the expression of ISGs [62,63,64,65,66]. Eventually, these ISGs perform antiviral functions, and therefore host establishes the antiviral status [59,67,68]. In addition, IFN affects several other processes, including cell growth, differentiation, and apoptosis, as well as immune regulation [69,70,71].

IFN and IFN-induced ISGs are so important for the maintenance of host antiviral status that their production undergoes complex and delicate regulation. ADP-ribosylation plays an important role in this process. Here, we first focus on ADP-ribosylation modification that occurs on proteins associated with IFN-I production and IFN-I-induced JAK-STAT signaling pathway (Figure 1). These include several proteins such as Mitochondrial antiviral signaling protein (MAVS), TANK-binding kinase 1 (TBK1) and STAT1. In addition, lots of ADP-ribosylation that occurs on other host antiviral factors can also affect host antiviral response. For example, PARP11 MARylates β-transducin repeat-containing protein (β-TrCP) and promote IFNAR1 ubiquitination and degradation, resulting in the downregulation of IFN-I signaling and antiviral activity. Next, we will describe these contents in detail.

### 2.1. ADP-Ribosylation of Proteins Associated with the Signaling Pathway of IFN-I Production

MAVS (also known as virus-induced signal adaptor [VISA]) is a key signaling molecule that mediates antiviral innate immune response initiated by RNA viruses [72,73,74,75]. In this process, retinoic acid-induced gene I (RIG-I) acts as a pattern recognition receptor to recognize and bind viral RNA in response to RNA virus infection [76]. Subsequent conformational change of RIG-I exposes its N-terminal caspase recruitment domain (2CARD), which binds to the N-terminal CARD domain of MAVS and induces MAVS polymerization at the mitochondrial outer membrane [77]. Active MAVS polymers recruit the tumor necrosis factor receptor-associated factor (TRAF) family to synthesize polyubiquitin chains that activate TBK1 and IκB-kinase (IKK). Activated TBK1 and IKK first phosphorylate MAVS so that phosphorylated MAVS can recruit IFN-regulatory factor 3 (IRF3) by binding to its conserved positively charged surface. When TBK1 and IRF3 are in close proximity, IRF3 is phosphorylated by TBK1. Finally, phosphorylated IRF3 dissociates from MAVS and translocates to the nucleus after forming a dimer, where it binds to the promoter of the IFN-I gene and drives type I IFN production [50]. Therefore, the strict and subtle dynamic regulation of MAVS is very important to antiviral immune response. PARP5a/Tankyrase 1 (TNKS1) and its homolog PARP5b/TNKS2 are known to catalyze the PARylation of their substrates. The five ankyrin repeat (ANK) units at the N-terminus of TNKS1 and TNKS2 are the structural basis for their substrate recognition. The C-terminal catalytic domain is responsible for the ADP-ribosylation of their substrates [78]. A recent study showed that TNKS1 and TNKS2 can poly-ADP-ribosylate MAVS at Glu137 residue. After viral infection, TNKS1 and TNKS2 are upregulated and translocate from the cytosol to mitochondria, interacting with MAVS and catalyzing its PARylation [36]. The PARylation of MAVS serves as a signal for the ubiquitin E3 ligase Ring figure protein 146 (RNF146)-mediated K48-linked polyubiquitination and subsequent degradation of MAVS [79], thereby negatively regulating the innate immune response to RNA viruses.

As mentioned above, TBK1 is a key kinase that induces IFN-I production. PARP7 (TIPARP) is an ADP-ribosylase whose expression is upregulated by aryl hydrocarbon receptor (AHR) [38,80]. AHR is a ligand-activated transcription factor that can be activated by a variety of environmental xenobiotics. Therefore, there is a close connection between AHR and innate immune signaling [80,81,82,83]. During viral infection, AHR-induced PARP7 can interact with TBK1 and catalyze its mono-ADP-ribosylation, which suppresses the activation of TBK1 and subsequent phosphorylation of IRF3. Consistent with this, AHR-deficient (*Ahr^−/−^*) MEFs and MEFs with PARP7-knockdown by siPARP7 show stronger antiviral effects [38]. Thus, the AHR-PARP7 axis is a negative regulator of interferon signaling. However, the ADP ribosylation site of TBK1 remains to be determined. It is reported that PARP7 mainly modifies cysteines and acidic amino acids (glutamates and aspartates), which also provides a basis for the determination of the ADP-ribosylation site of TBK1 [84,85].

### 2.2. ADP-Ribosylation of Proteins Associated with IFN-I-Induced JAK-STAT Signaling Pathway

STAT1 stands out as a key functional component of the interferon signaling pathway, and its post-translational modification profoundly affects signal transduction [86]. For example, linear ubiquitination of STAT1 inhibits STAT1 activation and thus lowers the strength of IFN-I antiviral signaling [66]. The ADP-ribosylation of STAT1 was first reported by Iwata et al. [40]. It is PARP14 that catalyzes mono-ADP-ribosylation of STAT1 and prevents its phosphorylation. Consistent with this, PARP14 silencing and STAT1 ribosylation site mutation promote STAT1 phosphorylation and STAT1-driven ISG expression, thus enhancing interferon signaling. Interestingly, PARP9 and PARP14 physically and functionally interact with each other. Protein ribosylation assay and mass spectrometry showed that PARP9 suppresses PARP14-mediated MARylation of STAT1, which sustains STAT1 phosphorylation. In other words, PARP14 and PARP9 play opposite roles in this process [40]. However, STAT1 SUMOylation also affects its phosphorylation, which in turn affects interferon downstream events [87]. The ADP-ribosylation of STAT1 has therefore been questioned and requires further evidence. It is worth noting that in mouse bone marrow-derived macrophages (BMDMs), the mRNA level of PARP14 is significantly upregulated when stimulated by IFNβ or toll-like receptor (TLR) agonists such as polyinosinic-polycytidylic acid (Poly (I: C)). Depletion of PARP14 results in a decrease in IRF3-mediated IFNβ production and ISG expression, thereby impairing the interferon response [88]. Taken together, PARP14 plays a complex role in the process of interferon resistance to pathogens.

### 2.3. ADP-Ribosylation of Other Host Antiviral Factors

β-TrCP is the ubiquitin E3 ligase of interferon-alpha/beta receptor 1 (IFNAR1) and mediates IFNAR1 ubiquitination and degradation [89]. β-TrCP belongs to the F-box/WD40 repeats family, which contains an F-box motif and seven WD40 motifs [90]. Among them, WD40 repeats are responsible for the binding of β-TrCP to its protein substrate. PARP11 can mono-ADP-ribosylate β-TrCP in the WD40 repeats. The MARylatopn of β-TrCP inhibits its ubiquitin-proteasome degradation and enhances the ability of β-TrCP to interact with IFNAR1, which in turn promotes IFNAR1 ubiquitination and degradation. The above process ultimately results in the downregulation of IFN-I signaling and antiviral activity. Moreover, viral infection could lead to the upregulation of PARP11, thus restricting IFN-I-induced expression of ISGs and promoting ADP-ribosylation-mediated immune evasion of the virus. Similarly, PARP11 knockdown significantly downregulates the protein level of β-TrCP and upregulates the protein level of IFNAR1, therefore enhancing the IFN-I-activated signaling pathway and antiviral activity. Taken together, PARP11 and PARP11-mediate-MARylation are highly effective targets for improving the antiviral efficacy of type-I interferon [42]. Consistent with this, a recent study shows that during Influenza A viruses (IAV) infection, PARP1 activity could facilitate IAV-induced IFNAR1 degradation, thus promoting virus propagation [35]. However, the substrate for this ADP-ribosylation process remains to be determined.

Initially, PARP9 was also thought to be enzymatically inactive. PARP9 was reported to form a heterodimer complex with Deltex E3 ubiquitin ligase 3L (DTX3L) [91]. DTX3L and PARP9 are relatively highly expressed in prostate cancer and breast cancer and share a common promoter. Both genes are responsive to IFNγ and activated in cells that express a dominant active form of STAT1 [92]. In addition, the complex can interact with STAT1 to promote STAT1-mediated ISG expression and thus enhance antiviral response [93]. Interestingly, in 2017, a study found that PARP9 can display mono-ADP-ribosylation activity in the case of DTX3L as its chaperone. DTX3L is an E3 ligase containing a RING domain, and PARP9/DTX3L heterodimer still has an E3 ligase activity. In the presence of high levels of NAD^+^, E1 and E2 enzymes, PARP9/DTX3L heterodimer exhibits ADP-ribosyltransferase activity and catalyzes mono-ADP-ribosylation of ubiquitin (Ub) at Gly76, which is involved in Ub conjugation to substrates. Therefore, the ADP-ribosylation of the Ub restricts the E3 function of DTX3L. In this process, both the RING domain of DTX3L and the catalytic domain of PARP9 are indispensable [39]. Although it has been reported that PARP9/DTX3L can target host histone H2BJ to enhance ISG expression and target encephalomyocarditis virus (EMCV) 3C protease to disrupt viral assembly as an E3 ubiquitin ligase [93], current evidence shows that PARP9 can only ADP-ribosylate Ub. It highlights the selectivity of ADP-ribosylation. However, the effect of PARP9-mediated MARylation on the antiviral activity of PARP9/DTX3L remains to be further investigated.

In addition, PARP7 can MARylate other proteins that function in antiviral immunity, such as PARP13. Recent studies have shown that PARP7 MARylates PARP13 at cysteines (Cys), rather than glutamates (Glu) or aspartates (Asp) [23]. Of course, there are many examples of PARP7 modifying these two amino acids, such as PARP7 MARylates α-tubulin at glutamic acid and aspartic acid residues [84]. PARP13 is also a member of the PARP family, which is considered to be catalytically inactive [13]. Even so, PARP13 is still known for its antiviral activity [48]. PARP13 has shown antiviral activity against a variety of DNA and RNA viruses, including influenza A virus, alphaviruses, filoviruses, and human immunodeficiency virus 1 (HIV-1) [45,46,47,94]. Mechanistically, PARP13 can promote interferon signaling by interacting with RIG-I and other ISGs. In addition, PARP13 can also target viral RNA to induce its degradation [95,96,97]. Since the Cys residues in the zinc-finger domains of PARP13 are responsible for the coordination of Zn^2+^ [23,85], the MARylation of these Cys residues may prevent RNA binding. Thus, the Cys MARylation of PARP13 appears to limit the antiviral response. Of note is that the MARylation of Cys in cells is more stable than Glu/Asp, suggesting that the MARylation site of protein targets regulates the duration of the signal. Moreover, PARP14 can also MARylate PARP13 on several Glu/Asp residues, providing a link for crosstalk among PARP family members [49].

## 3. ADP-Ribosylation of Virus-Encoded Proteins

Recent studies have found that viral proteins can also undergo ADP-ribosylation modification. These include the Epstein-Barr nuclear antigen 1 (EBNA1) of Epstein-Barr virus (EBV), the latency-associated nuclear antigen (LANA) of Kaposi’s sarcoma-associated herpesvirus (KSHV), the nonstructural proteins NS1 and NS3 of Zika virus (ZIKV), the nonstructural polyprotein nsP2 of Chikungunya virus (CHIKV), the nucleocapsid (N) protein of coronavirus (CoVs) and the core proteins of adenovirus.

EBV is a human herpesvirus, and EBNA1 is the only viral protein required for the stable maintenance of the viral genome [98,99]. It is reported that PARP1 and TNKS can interact directly with the EBNA1 protein and catalyze its poly-ADP-ribosylation, resulting in the inhibition of OriP replication [33,37]. In the same way, PARP1 can also PARylate LANA of KSHV. KSHV, another human herpesvirus, is usually in the latent phase of disease as reported for EVB [34,100,101]. In latent replication, the KSHV genome is considered to replicate once per cell cycle, and several potentially pathogenic genes such as LANA, K-cyclin (ORF72), K15, and vFLIP (ORF71) are expressed [102]. Thus, the poly-ADP-ribosylation of LANA appears to affect the maintenance of the viral genome. Taking the above discussion together, PARP1- and TNKS-mediated ADP-ribosylation of viral proteins inhibit viral replication and infection using a similar mechanism.

ZiKV is a mosquito-borne flavivirus with a single-stranded positive RNA genome [103,104], and its nonstructural viral proteins NS1 and NS3 have been reported to be targeted by PARP12 [44]. In this process, PARP12 is responsible for the initial MARylation reaction of NS1 and NS3. Subsequent PARylation is performed by other PARPs. This modification of NS1 and NS3 triggers their K48-linked ubiquitination and proteasome-mediated degradation. Since NS1 and NS3 are involved in viral replication and immune evasion [105,106], the PARP-dependent degradation of NS1 and NS3 suppresses Zika virus infection and immune evasion [44]. In addition, a recent study finds that PARP11 can enhance PARP12-mediated NS1 and NS3 ADP-ribosylation and degradation, thus inhibiting Zika virus infection [43]. However, the exact mechanism remains to be elucidated. Again, the ADP-ribosylation of ZiKV proteins shows the potent antiviral activity of this modification.

CHIKV is a mosquito-borne virus with a genome of approximately 11,800 nucleotides [107]. Early in infection, CHIKV encodes a nonstructural polyprotein (nsP). Subsequently, nsP is cleaved into four separate nsP1–4 that together form a replication complex. In this process, nsP2 functions as a key protease. Therefore, it is an important target for antiviral drugs [108,109,110,111]. PARP10 can MARylate nsP2 and its protease domain and inhibit its proteolytic activity, thus restricting the processing of nsP and suppressing CHIKV replication. Interestingly, nsP3 is responsible for MAR hydrolase activity and can remove the MARylation modification from nsP2, thereby reactivating its proteolytic activity and promoting CHIKV infection [41,112,113]. In summary, ADP-ribosylation plays an indispensable role in host antiviral response.

In addition, it has been reported that some viral proteins can undergo ADP-ribosylation, but which PARP catalyzes this process, and the effect of ADP-ribosylation on viral infection are still unclear. For example, the coronavirus (CoVs) nucleocapsid (N) protein is ADP-ribosylated in cells during coronavirus infection, and the nsp3 macrodomain does not affect ADP-ribosylation of the N protein. N protein ADP-ribosylation can only be detected in the context of viral infection and cannot be detected in mock-infected cells. Interestingly, nucleocapsid protein ADP-ribosylation is conserved in both α- and β-coronaviruses [114]. This suggests that it is important for viral replication in the host or host resistance to the virus. Future experiments are needed to explore the function of N protein ADP-ribosylation. Another example is that adenovirus core proteins also undergo ADP-ribosylation during viral infection and may play a role in virus decapsidation [115]. However, more details remain to be explored.

## 4. ADP-Ribosylation of Nucleic Acid Molecules

It is of interest to mention that in addition to proteins, many nucleic acid molecules can also undergo MARylation or PARylation modification [116,117,118,119,120], but we still know relatively little about them. The first nucleic acid discovered to undergo ADP-ribosylation was bacterial DNA, which results in an inhibition of bacterial DNA replication. The antitoxin DarG, a macrodomain protein, catalyzes this process [121]. This suggests that ADP-ribosylation of nucleic acids may also be present in mammals. In fact, it has been shown that PARP1 and PARP2 can PARylate the phosphorylated ends of double-stranded or single-stranded DNA in vitro [122,123]. Similarly, PARP3 can MARylate DNA substrates [124]. In addition, RNA can also serve as a substrate for ADP-ribosylation. In vitro experiments showed that PARP10 can modify the 5′-phosphorylated termini of RNA [119]. Post-translational modifications of these nucleic acid molecules provide new insights into the molecular mechanisms of ADP-ribosylation modifications mediated by PARPs. However, these in vitro experiments still need more validation, especially in vivo validation. We do not yet know whether PARPs are also sequence-specific when recognizing nucleic acid molecules, which still requires more exploration. The next step is to explore whether ADP-ribosylation occurs on viral genomes and whether it is related to antiviral immunity response. This might provide a seed for subsequent studies.

## 5. Conclusions and Perspectives

As discussed above, ADP-ribosylation plays a complex function in antiviral response. ADP-ribosylation of proteins involved in IFN-I signaling pathways generally weakens antiviral responses, whereas inhibition of this process enhances antiviral signaling. Interestingly, several of the rapidly evolving PARP genes, including PARP9, PARP10, PARP12, PARP13, and PARP14, are upregulated by IFN, and overexpression of them upregulates several antiviral effectors. Therefore, these PARPs are considered as ISGs [125,126,127,128]. ADP-ribosylation of viral proteins often promotes their degradation and thus enhances the host antiviral response, while viruses have also evolved strategies to reverse ADP-ribosylation modifications. One common strategy is that viral macrodomains can degrade ADP-ribosylation modification. As previously described, the nsP3 macrodomain of CHIKV is able to remove the MARylation modification of nsP2. In addition, severe acute respiratory syndrome coronavirus (SARS-CoV) macrodomain mutations significantly increase the virus’s sensitivity to interferon. Similarly, mutations in the ADP-ribose binding region of the Sindbis virus (SINV) macrodomain impair viral replication [129,130]. All these suggest that viral macrodomains may play an important role in antiviral immune escape. It appears that viral macrodomains are phylogenetically and structurally closely related to cellular macrodomains. It is, therefore, not surprising that viral macrodomains can hydrolyze ADP-ribosylation [41,131,132,133]. Together, these observations offer mechanisms of how ADP-ribosylation functions in host-virus conflicts. It seems that ADP-ribosylation can both enhance and restrict the antiviral response, depending on the specificity of the substrate. Therefore, a thorough understanding of the role of ADP-ribosylation in the antiviral response is crucial for the prevention and treatment of virus-associated diseases.

ADP-ribosylation plays such an important role in many biological processes, such as innate immunity, that PARP inhibitors (PARPi) have also become a research hotspot in recent years. For example, rucaparib inhibits PARP11-induced β-TrCP ADP-ribosylation, leading to a decrease in β-TrCP level and an increase in IFNAR1 level [42]. Therefore, rucaparib can effectively promote type-I IFN signaling pathway transduction and enhance host antiviral activity. Other PARP inhibitors, such as olaparib, veliparib, and niraparib, are still under experimental or clinical investigation with rapid progress [134,135]. However, there is still a long way to go in the development of PARP inhibitors due to the difficulty of developing specific inhibitors for individual PARPs and the drug resistance of PARP inhibitors.

Despite these notions, we would like to point out that only a few direct substrates of ADP-ribosylation have been identified so far. Thus far, we do not have a deep understanding of how PARPs select and interact with target substrates and how they are activated [116]. It might be interesting to define more broadly relevant substrates (both host and viral factors), which requires further exploration. Certainly, the identification of specific sites of ADP-ribosylation is equally important. With the development of mass spectrometry techniques and detection tools, it is possible to identify specific substrates for ADP-ribosylation and map modification sites. This will contribute to a further understanding of the role of ADP-ribosylation in antiviral response.

## Figures and Tables

**Figure 1 pathogens-12-00303-f001:**
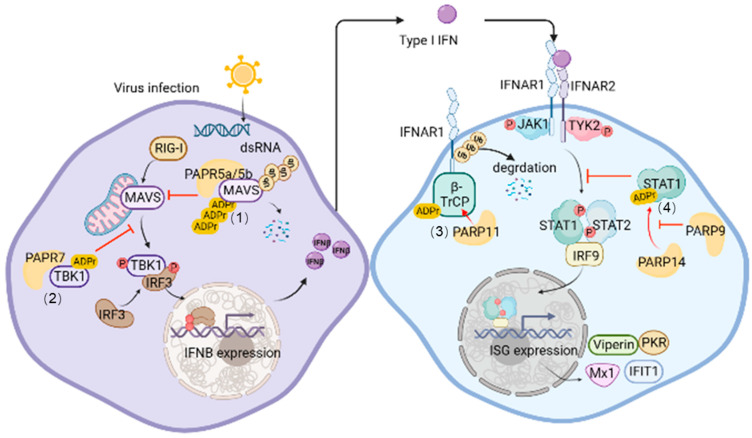
ADP-ribosylation in IFN-I production and IFN-I-induced JAK-STAT signaling pathway. During viral infection, (1) TNKS1 and TNKS2 interact with MAVS and catalyze its PARylation, leading to the degradation of MAVS and impairment of IFN production. (2) PARP7 targets TBK1 and catalyzes it mono-ADP-ribosylation, which suppresses the activation of TBK1 and downstream events. (3) PARP11 mono-ADP-ribosylates β-TrCP and mediates IFNAR1 ubiquitination and degradation, thus acting as a negative regulator of IFN-I response. (4) PARP14 catalyzes mono-ADP-ribosylation of STAT1 and prevents its phosphorylation, thus repressing the IFN-I response. PARP9 suppresses PARP14-mediated MARylation of STAT1, which sustains STAT1 phosphorylation. (Created with BioRender.com (accessed on 16 January 2023).

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
