# Peer review of "ADP-Ribosylation in Antiviral Innate Immune Response"

_pathogens, 2023, doi:10.3390/pathogens12020303_

Round 1

Reviewer 1 Report

This is a very timely article on this interesting area, clearly laying out the issues. It is well-written and will be used by researchers interested in the field of antiviral innate immunity response. 

Suggestions for improvement:

1) Table I: When no roles have been discovered, add "not known" in the corresponding column. In addition, references should also be cited. As for the role of PARP7, please be more precise by replacing "targeting" with "inhibiting";  Roles: MARylating and inhibiting TBK1.

2) Lanes 66 and 67: to write that the roles of type III IFN are poorly understood is somehow misleading. Several recent studies and reviews are available describing their roles in mucosal immunity.

3) Lanes 103 and 104: I am not convinced that AHR can be classified, or considered as PRR. 

4) Figure 1 is too small.

Reviewer 2 Report

This paper describes the ADP-ribosylation in antiviral innate response.

In human, there are 17 PARPs, which plays essential roles in many biological processes, such as DNA damage repair, gene regulation, and energy metabolism. The author summarizes the ADP-ribosylation that occurs on both host and viral proteins and its roles.

The review is well summarized in this field over the last years. Especially, PARP enzymes are broad diversity, and its substrate has been not clearly defined. Furthermore, their cross-talk and reversible regulation are much more complex. Therefore, to understand the field, I think it will be good review.

Minor points

L110 Iwata et al> ref 62

L114 PARP9 suppresses PARP14-mediated MARylation of STAT1

Please add more clear explanation of how PARP9 suppresses PARP14-mediated MARylation.

L144~L146 PARP9 can display mono-ADP-ribosylation activity in the case of DTX3L against ubiquitin. This is an interesting topic.

As far as I know, the only two members, PARP9 and PARP13, do not share the H-Y-Y-E motif, and are generally regarded as catalytically inactive.

Please add an explanation of how PARP9 exerts ADP-ribosylation activity because PARP9 lacks the histidine on the active site (PARP9 Q, PARP13 Y). 

Table1 > PARP activity inactive PARP13 only? 

As mentioned above, it may be better to add information about DTX3L.

L208 retains its proteolytic activity > no activity? 

Otherwise, please add a precise explanation.
